# Coffee consumption and risk of breast cancer: A Mendelian randomization study

**Merete Ellingjord-Dale**[1]*, **Nikos Papadimitriou**[2], **Michail Katsoulis**[3], **Chew Yee**[1], **Niki Dimou**[2], **Dipender Gill**[1], **Dagfinn Aune**[1,4,5], **Jue-Sheng Ong**[6,7], **Stuart MacGregor**[6], **Benjamin Elsworth**[7], **Sarah J. Lewis**[7,8], **Richard M. Martin**[7,8,9], **Elio Riboli**[1], **Konstantinos K. Tsilidis**[1,10]

1 Department of Epidemiology and Biostatistics, School of Public Health, Imperial College London, London, United Kingdom, 2 Section of Nutrition and Metabolism, International Agency for Research on Cancer, Lyon, France, 3 Institute of Health Informatics Research, Faculty of Population Health Sciences, University College London, London, United Kingdom, 4 Department of Nutrition, Bjørknes University College, Oslo, Norway, 5 Department of Endocrinology, Morbid Obesity and Preventive Medicine, Oslo University Hospital, Oslo, Norway, 6 Statistical Genetics, Department of Genetics and Computational Biology, QIMR Berghofer Medical Research Institute, Brisbane, QLD, Australia, 7 MRC Integrative Epidemiology Unit, University of Bristol, Bristol, United Kingdom, 8 Department of Population Health Sciences, Bristol Medical School, University of Bristol, Bristol, United Kingdom, 9 National Institute for Health Research (NIHR) Bristol Biomedical Research Centre, University Hospitals Bristol NHS Foundation Trust and the University of Bristol, Bristol, United Kingdom, 10 Department of Hygiene and Epidemiology, School of Medicine, University of Ioannina, Ioannina, Greece

* m.ellingjord-dale@imperial.ac.uk

**Data Availability Statement:** All relevant data are within the manuscript and its Supporting Information files.

## Abstract

### Background

Observational studies have reported either null or weak protective associations for coffee consumption and risk of breast cancer.

### Methods

We conducted a two-sample Mendelian randomization (MR) analysis to evaluate the relationship between coffee consumption and breast cancer risk using 33 single-nucleotide polymorphisms (SNPs) associated with coffee consumption from a genome-wide association (GWA) study on 212,119 female UK Biobank participants of White British ancestry. Risk estimates for breast cancer were retrieved from publicly available GWA summary statistics from the Breast Cancer Association Consortium (BCAC) on 122,977 cases (of which 69,501 were estrogen receptor (ER)-positive, 21,468 ER-negative) and 105,974 controls of European ancestry. Random-effects inverse variance weighted (IVW) MR analyses were performed along with several sensitivity analyses to assess the impact of potential MR assumption violations.

### Results

One cup per day increase in genetically predicted coffee consumption in women was not associated with risk of total (IVW random-effects; odds ratio (OR): 0.91, 95% confidence intervals (CI): 0.80–1.02, P: 0.12, P for instrument heterogeneity: 7.17e-13), ER-positive

**Funding:** This work was supported by the World Cancer Research Fund International Regular Grant Programme (WCRF 2014/1180 to Konstantinos K. Tsilidis). RMM was supported by a Cancer Research UK (C18281/A19169) programme grant (the Integrative Cancer Epidemiology Programme) and is part of the Medical Research Council Integrative Epidemiology Unit at the University of Bristol supported by the Medical Research Council (MC_UU_12013/1, MC_UU_12013/2, and MC_UU_12013/3) and the University of Bristol. RMM is also supported by the National Institute for Health Research (NIHR) Bristol Biomedical Research Centre which is funded by the National Institute for Health Research (NIHR) and is a partnership between University Hospitals Bristol NHS Foundation Trust and the University of Bristol. The funders had no role in study design, data collection and analysis, decision to publish, or preparation of the manuscript.

**Competing interests:** The authors have declared that no competing interests exist.

**Abbreviations:** BMI, body mass index; BCAC, the Breast Cancer Association Consortium; CI, Confidence interval; ER, estrogen receptor; GWA, Genome-wide association; IVW, Inverse variance weighted; MR, Mendelian randomization; OR, Odds ratio; SNP, single-nucleotide polymorphism; UKB, UK Biobank.

(OR = 0.90, 95% CI: 0.79–1.02, P: 0.09) and ER-negative breast cancer (OR: 0.88, 95% CI: 0.75–1.03, P: 0.12). Null associations were also found in the sensitivity analyses using MR-Egger (total breast cancer; OR: 1.00, 95% CI: 0.80–1.25), weighted median (OR: 0.97, 95% CI: 0.89–1.05) and weighted mode (OR: 1.00, CI: 0.93–1.07).

## Conclusions

The results of this large MR study do not support an association of genetically predicted coffee consumption on breast cancer risk, but we cannot rule out existence of a weak association.

## Background

Coffee contains biochemical compounds such as caffeine, polyphenols and diterpenes that may protect against breast cancer risk through their anticarcinogenic properties [1–3] or through their favorable alterations of levels of estradiol and SHBG [4–8]. Several observational studies have investigated the association between coffee consumption and breast cancer risk, but findings have been inconsistent with the majority of studies reporting null associations [9–25] and other studies reporting protective associations [26–30]. A recent meta-analysis including 21 prospective cohort studies reported a weak protective association for highest versus lowest category of coffee consumption with overall (RR = 0.96, 95% CI = 0.93–1.00) and postmenopausal (RR = 0.92, 95% CI = 0.88–0.98) breast cancer [31]. However, observational studies may be confounded by other dietary or lifestyle factors. Further, there are no clinical trials on the effect of coffee consumption on breast cancer risk, and it is still unclear whether an association exists and if so, whether it is causal.

Several genome-wide association studies (GWAS) on coffee or caffeine consumption have been previously published [32–37]. One of these GWAS was a meta-analysis conducted by the Coffee and Caffeine Genetics Consortium in 2015 incorporating summary statistics from 28 population-based studies of European ancestry, and reported six loci associated with coffee consumption that were involved either in the pharmacokinetics (cytochrome P4501A1 (CYP1A1)/cytochrome P4501A2 (CYP1A2), aryl hydrocarbon receptor (AHR)) or pharmacodynamics of caffeine (brain-derived neurotrophic factor (BDNF) and solute carrier family 6 member 4 (SLC6A4)) [35]. A more recent and larger GWAS was conducted among individuals (179,954 males and 212,119 females) of white British ancestry in the UK Biobank (UKB) cohort [37], and identified 35 genetic variants strongly associated with coffee intake.

Mendelian randomization (MR) is a method that uses genetic variation arising from meiosis as a natural experiment, to investigate the potential causal relationship between an exposure and an outcome [38, 39]. MR estimates are less susceptible to bias from potential reverse causality and confounding compared to estimates from observational studies, because genetic variants are randomly distributed at conception [40, 41]. A recent MR study assessed the potential causal association between coffee consumption and risk of several cancers, including breast cancer, and concluded that coffee consumption is unlikely to be associated with overall breast cancer susceptibility [37]. However, the latter study did not report associations by breast cancer subtypes. In the current MR study, we investigated the relationship between genetically predicted coffee consumption and risk of breast cancer overall as well as breast cancer subtypes incorporating several MR methods to assess the impact of potential MR assumption violations.

## Methods

### Genetic data on coffee consumption

We used 35 single nucleotide polymorphisms (SNPs) that were associated with coffee consumption at genome-wide significance (p<5e-8) level in the combined population of men and women in UKB [37], but their beta estimates (SNP-coffee) were derived from analyses only among the female population. In a sensitivity analysis, we combined beta estimates (SNP-coffee) for both men and women to increase statistical power. The UKB is a population-based cohort study of more than 500,000 participants aged 38 to 73 years, who enrolled in the study between 2006 and 2010 from across the UK [42]. Coffee consumption was measured via self-administered questionnaires and was defined as cups of decaffeinated coffee, instant coffee, ground coffee and any other type of coffee (UKB Data field ID: 1508) consumed per day [37]. Briefly, the UKB participants were genotyped using Affymetrix UK Biobank Axiom array and imputed against the UK10K, 1000 Genomes Phase 3 and Haplotype Reference Consortium panels [37]. The GWAS was conducted using the BOLT-LMM software [43] to model the genetic association accounting for cryptic relatedness in the UKB sample. SNPs were clumped at $r^2$ <0.01 using a 10-mb window [37].

### Genetic data on breast cancer

Out of the 35 genome-wide significant SNPs [37], we extracted 33 SNPs from the publicly available breast cancer GWAS from the Breast Cancer Association Consortium (BCAC). BCAC has data on 122,977 breast cancer cases of which 69,501 were estrogen receptor (ER)-positive, 21,468 ER-negative, and 105,974 controls of European ancestry (http://bcac.ccge. medschl.cam.ac.uk/bcacdata/oncoarray/gwas-icogs-and-oncoarray-summary-results/). BCAC was initiated in 2005 and is an international collaboration that studies genetic susceptibility to breast cancer. The breast cancer GWAS was performed in females of European ancestry from 68 studies collaborating in BCAC, the Discovery, Biology and Risk of Inherited Variants in Breast Cancer Consortium (DRIVE; 61,282 cases and 45,494 controls), the Illumina iSelect genotyping Collaborative Oncological Gene-Environment Study (iCOGS; 46,785 cases and 42,892 controls), and 11 other breast cancer GWAS (14,910 cases and 17,588 controls) [44]. Genotyping in the BCAC and DRIVE studies was done using OncoArray1, whereas iCOGS used Illumina iSelect array (http://ccge.medschl.cam.ac.uk/research/consortia/icogs/). Using the 1000 Genomes Project (Phase 3) reference panel, genotypes were imputed for approximately 21M variants [44].

### Statistical power

Statistical power calculations were conducted using the online mRnd calculator (available at http://cnsgenomics.com/shiny/mRnd/). Using an estimated 1% variance of coffee consumption explained by the instruments [37], the study had 80% power with a type I error rate of 0.05 to detect associations of odds ratios of 0.89, 0.87 and 0.80 per one cup of coffee per day and risk of overall, ER-positive and ER-negative breast cancer, respectively.

### Statistical analysis

**Main MR analysis.** We conducted a two-sample MR using summary association data for 33 coffee-associated SNPs. We ran both fixed- and random-effects inverse-variance weighted (IVW) models, but the random-effects IVW model was considered the main analysis due to the large number of SNPs and the substantive observed heterogeneity [45, 46]. The IVW MR approach combines individual MR estimates across SNPs to derive an overall weighted

estimate of the potential causal effect. We calculated the MR-derived odds ratio (OR) of breast cancer risk for a one cup per day increase in genetically predicted coffee consumption. This study used publicly available data.

**Sensitivity analyses.**   The IVW MR approach assumes that all genetic variants must satisfy the instrumental variable assumptions, namely the genetic variants must be: 1) associated with coffee consumption, 2) not associated with confounders of the association between coffee consumption and breast cancer, and 3) only associated with breast cancer via their association with coffee consumption [45, 47, 48]. We tested for potential violation of the first MR assumption by measuring the strength of the genetic instruments using F-statistics. The F-statistic is the ratio of the mean square of the model to the mean square of error [49]. The Cochran's Q test and the $I^2$ statistic were used to quantify the heterogeneity in effect sizes between the genetic instruments [50], which may indicate horizontal pleiotropy that could violate the third MR assumption. To further test and attempt to correct for potential violation of the second and third MR assumptions, we used several approaches such as the MR-Egger regression [51], the weighted median [52] and mode [53] methods, and the MR pleiotropy residual sum and outlier test (MR-PRESSO) [54].

**MR-Egger.**   The MR-Egger is an adaption of Egger regression, which allows for directional pleiotropy by introducing an intercept in the weighted regression model. Values away from zero for the intercept term are an indication of horizontal pleiotropy [51]. The MR-Egger approach provides unbiased results in the presence of pleiotropic instruments assuming that the magnitude of pleiotropic effects is independent of the size of the SNP-coffee consumption effects, which is called the Instrument Strength Independent of Direct Effects (InSIDE) assumption [51].

**Weighted median.**   We used the weighted median method that orders the MR estimates obtained using each instrument weighted for the inverse of their variance. Selecting the median result provides a single MR estimate with confidence intervals estimated using a parametric bootstrap method [52]. The weighted median does not require that the size of any pleiotropic effects on the instruments are uncorrelated to their effects on the intermediate phenotype, but assumes that at least half of the instruments are valid [55].

**Weighted mode.**   The mode based causal estimate consistently estimates the true causal effect when the largest group of instruments with consistent MR estimates is valid [53].

**MR-PRESSO.**   We used the MR-PRESSO outlier test to identify outlier SNPs, which could have pleiotropic effects [54]. This method regresses SNP-outcome on SNP-exposure and uses square of residuals to identify outliers.

To further determine whether pleiotropy could have influenced our results, we collected information on published associations of the genetic instruments for coffee consumption with other phenotypes from the Phenoscanner webpage [56]. Genetic instruments associated at genome-wide significance with potentially important confounders of the coffee and breast cancer association, namely BMI [57–61], age at menarche [62, 63], alcohol [64–68], smoking [67, 69–71] and age at menopause [72] were iteratively excluded from the analyses.

In addition, we repeated the analysis after excluding SNPs that had p-values in their associations with coffee consumption among women larger than 1e-05 to avoid weak instrument bias. We also used beta estimates from a previous GWAS as an alternative instrument of eight SNPs (rs1260326, rs1481012, rs17685, rs7800944, rs6265, rs9902453, rs2472297 and rs4410790) associated with coffee consumption [35] to ensure that our results were robust against different choices of instrument selection and because these eight SNPs are linked to caffeine metabolism and may reflect less likelihood for pleiotropic actions. All the analyses were performed using the MR robust package in Stata [73] and the Mendelian randomization package in R [74].

## Results

The associations between the genetic instruments with coffee consumption and breast cancer are shown in the S1 Table. One variant (rs17817964 in *FTO*) was strongly associated with overall (P = 4.67E-20), ER-positive (P = 2.48E-13) and ER-negative breast cancer (P = 1.56E-09).

### Main MR analyses

The fixed-effects IVW method yielded inverse associations for genetically predicted coffee intake and risk of total, ER-positive and ER-negative breast cancer (Figs 1–3 and S2 Table), but there was substantial heterogeneity in the individual SNPs instrumenting coffee and risk of disease (Cochran's Q test P-value = $10^{-5}$–$10^{-13}$, $I^2$ = 57–74%, S1–S6 Figs). Therefore, the random-effects IVW model was preferentially adopted for the main analysis, where the association between coffee consumption (per cup of coffee per day) and total (OR = 0.91, 95% CI = 0.80–1.02, P = 0.12), ER-positive (OR = 0.90, 95% CI = 0.79–1.02, P = 0.09) and ER-negative breast cancer (OR = 0.88, 95% CI = 0.75–1.03, P = 0.12) resulted in wider confidence intervals overlapping the null (Figs 1–3 and S2 Table).

### MR-Egger

Results based on the MR-Egger regression did not show any association for genetically predicted coffee consumption and risk of total breast cancer or subtypes (Figs 1–3, S2 Table).

### Weighted median and mode

Similarly, results from the weighted median analysis showed little evidence of an association per one cup of coffee per day and overall (OR = 0.97, 95% CI = 0.89–1.05, P = 0.45, Fig 1), ER-positive (OR = 0.94, 95% CI = 0.86–1.04, P = 0.24, Fig 2) and ER-negative breast cancer (OR = 1.02, 95% CI = 0.90–1.17, P = 0.72, Fig 3). The weighted mode model also yielded little evidence for an association (Overall breast cancer; OR = 1.00, 95% CI = 0.93–1.07, Figs 1–3 and S2 Table).

### MR-PRESSO

The MR-PRESSO outlier test detected six SNPs as potential outliers for total breast cancer (i.e. rs13387939, rs17817964, rs34060476, rs2472297, rs2521501 and rs539515), three SNPs for ER-positive breast cancer (i.e. rs17817964, rs2472297 and rs2521501) and three SNPs for ER-negative breast cancer (i.e. rs13387939, rs3810291 and rs17817964). After excluding these SNPs outliers, there was an inverse association between genetically predicted coffee intake (per one cup of coffee per day) and risk of overall (OR = 0.90, 95% CI = 0.83–0.98, P = 0.03) and ER-positive breast cancer (OR = 0.87, 95% CI = 0.78–0.97, P = 0.02), but no association for ER-negative breast cancer (OR = 0.97, 95% CI = 0.87–1.08, P = 0.62, S2 Table). However, the rs2472297 is located between *CYP1A1* and *CYP1A2* and is involved in the pharmacokinetics of caffeine, and has the strongest association with coffee consumption amongst all genetic instruments (P< $1e^{-168}$). Many of the other outlying SNPs had genome-wide significant associations with age at menarche (rs17817964, rs13387939, rs539515 and rs3810291), body mass index (rs17817964, rs13387939, rs2472297, rs539515 and rs3810291) and alcohol intake (rs17817964 and rs34060476, S3 Table).

### Sensitivity analyses

We performed several sensitivity analyses and there was little evidence of any association between genetically predicted coffee consumption and breast cancer risk (S2 Table). We

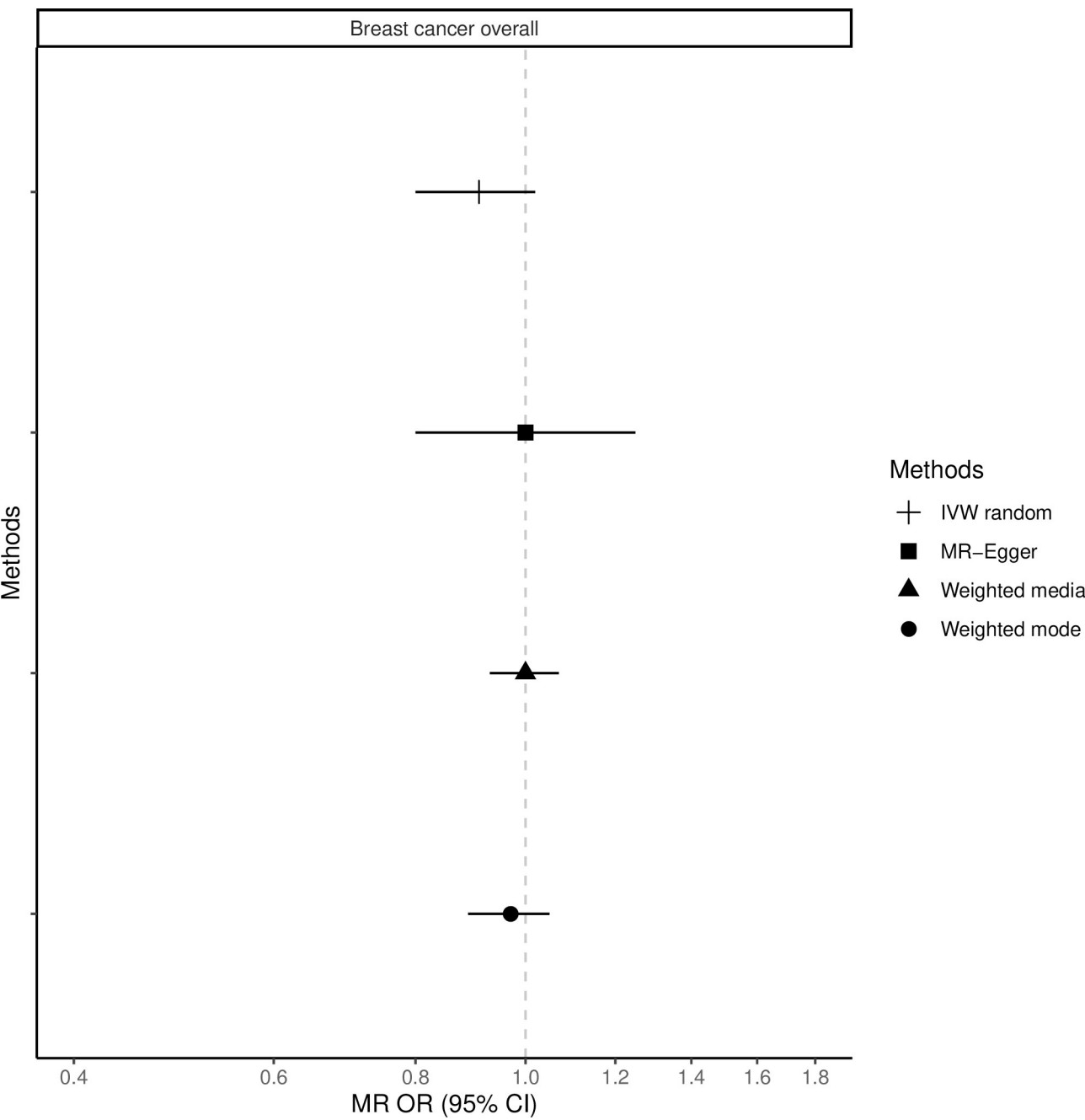

**Fig 1. Association between 1 cup/day increase of coffee consumption and breast cancer risk overall.** MR-analyses are derived using random effect IVW, MR-Egger, weighted median and mode.

performed MR-analyses after excluding genetic instruments known to be associated at genome-wide significance with 1) body mass index (i.e. rs4357572, rs539515, rs62106258, rs13387939, rs142219, rs2465054, rs4410790, rs2472297, rs17817964, rs66723169 and rs3810291), 2) age at menarche (i.e. rs539515, rs62106258, rs13387939, rs2236955, rs17817964 and rs381029), 3) alcohol consumption (i.e. rs1260326, rs34060476, rs17817964 and rs66723169), 4) smoking (i.e. rs56113850), and 5) age at menopause (i.e. rs1260326) (S2 and S3 Tables). When we reran the analyses after excluding 13 genetic instruments (i.e.

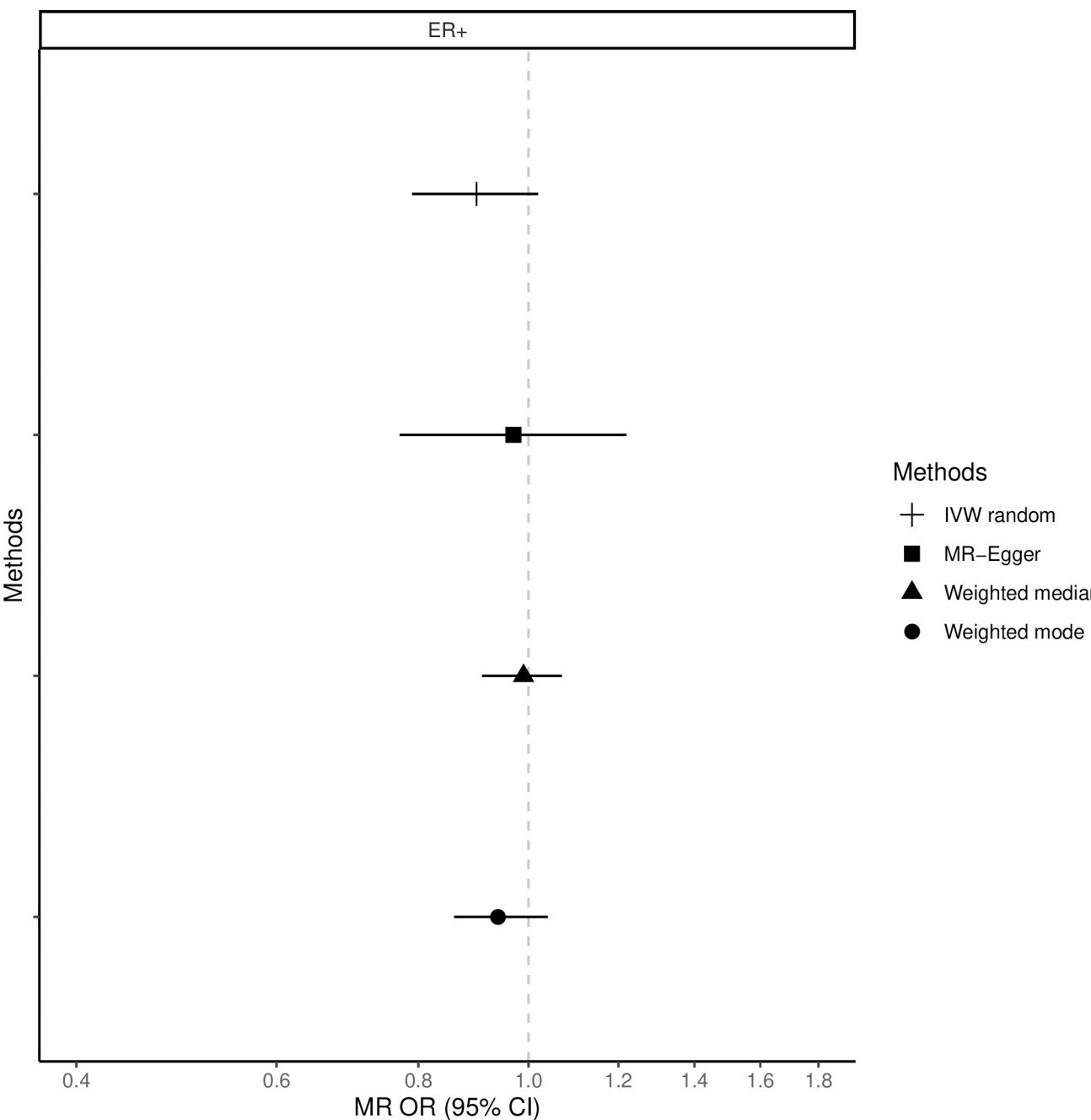

**Fig 2. Association between 1 cup/day increase of coffee consumption and risk of ER-positive breast cancer.** MR-analyses are derived using random effect IVW, MR-Egger, weighted median and mode.

rs117968677, rs1260326, rs1422191, rs16966903, rs2236955, rs2465054, rs2667773, rs34190000, rs3810291, rs395815, rs4092465, rs55754437 and rs62064918) with p-values with coffee consumption among women larger than $10^{-5}$, the results remained largely the same (Overall; OR = 0.90, 95% CI 0.77–1.06, ER-positive; OR = 0.99, 95% CI 0.77–1.05 and ER-negative; OR = 0.88, 95% CI 0.72–1.07, S2 Table). In another sensitivity analysis, we used as genetic instruments eight SNPs (i.e. rs1260326, rs1481012, rs17685, rs7800944, rs6265, rs9902453, rs4410790 and rs2472297) from a GWAS for coffee consumption among

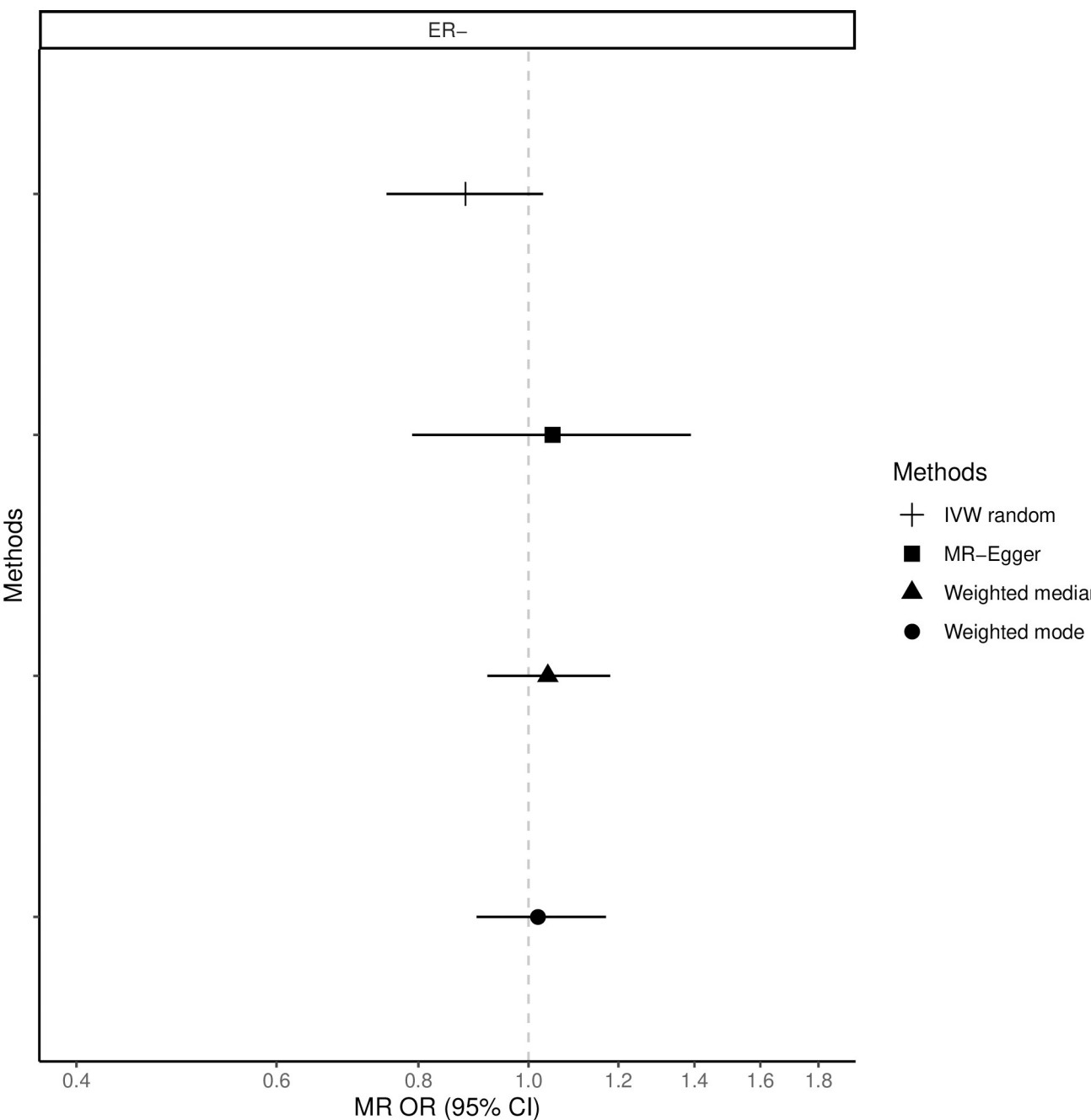

**Fig 3. Association between 1 cup/day increase of coffee consumption and risk of ER-negative breast cancer.** MR-analyses are derived using random effect IVW, MR-Egger, weighted median and mode.

consumers conducted by the Coffee and Caffeine Genetics Consortium [35], and there was again no evidence of an association (Overall; OR = 1.10, 95% CI 0.97–1.24, ER-positive; OR = 1.07, 95% CI 0.96–1.21 and ER-negative; OR = 1.16, 95% CI 0.97–1.38). To increase statistical power, we used the 33 genetic instruments from UK Biobank but with beta estimates (SNP-coffee) from females and males combined, but the results remained largely the same (Overall; OR = 0.92, 95% CI 0.82–1.04, P = 0.20, ER-positive; OR = 0.92, 95% CI 0.81–1.04, P = 0.16, ER-negative; OR = 0.90, 95% CI 0.77–1.05, P = 0.17, S2 Table).

## Discussion

In this comprehensive MR analysis of coffee consumption with risk of breast cancer, we observed that in the majority of analyses genetically predicted consumption of coffee was not associated with overall, ER-positive and ER-negative breast cancer. In line with our results, a recent large MR-study on the association between coffee consumption and risk of being diagnosed with or dying from cancer overall and by anatomical subsite reported no evidence for an association with risk of breast cancer [37]. Compared to the previous study, our study added results by ER-status and presented detailed sensitivity analyses to fully assess potential violations of MR assumptions.

Coffee is among the most commonly consumed beverages worldwide, and its drinking provides exposure to a range of biologically active compounds [75]. Higher coffee consumption has been associated with decreased risk of all-cause, cardiovascular and cancer mortality among non-smokers [76]. Several observational studies have investigated the association between coffee consumption and risk of breast cancer development, but findings have been inconsistent [31, 77, 78]. The most recent meta-analysis synthesized evidence from 21 prospective cohort studies [31], and reported a weak inverse association between coffee consumption and risk of total (OR higher vs. lower = 0.96, 95% CI = 0.93–1.00) and postmenopausal breast cancer (OR = 0.92, 95% CI = 0.88–0.98). Null associations were reported by estrogen or progesterone receptor status [31]. When a dose-response meta-analysis was conducted among 13 prospective studies [31], the association per one cup of coffee per day was nominally significant (OR for postmenopausal disease = 0.97, 95% CI = 0.95–1.00), which was consistent with the finding of the current MR study (OR = 0.90, 95% CI 0.79–1.02). In agreement, the World Cancer Research Fund Third Expert Report graded the evidence of coffee consumption and breast cancer risk as limited-no conclusion [79].

MR studies can be useful in nutritional epidemiology, as they are less susceptible to biases that are commonly present in traditional observational literature [80], namely exposure measurement error, residual confounding and reverse causation. MR estimates warrant a causal interpretation only if the assumptions of the instrumental variable approach hold. Though it is not possible to prove the validity of the assumptions in entirety, we performed several sensitivity analyses to detect potential violations and derived estimates that are potentially robust against violations of these assumptions. The majority of the sensitivity analyses supported our main analysis finding.

Several limitations should be considered when interpreting our findings. Our MR-analysis had appropriate statistical power to detect an OR of 0.89 per cup of coffee per day and risk of overall breast cancer. Observational studies have detected smaller associations of coffee consumption and breast cancer risk than this [31]. We were unable to rule out the possibility that coffee consumption may have a weaker association that we were not powered to detect. A weakness of using summary level data in two-sample MR is that stratified analyses by covariates of interest (e.g. smoking, alcohol, obesity, physical activity) are not possible which would have allowed us to investigate potential interactions between risk factors, but previous observational studies have in general not identified interactions with these variables [31]. Although we have involved clinically meaningful disease subtypes such as ER+ /− breast cancer, we could not examine breast cancer based on menopause status but 85% of breast cancer cases in our sample are postmenopausal. Although our genetic instruments are robustly associated with coffee consumption, coffee consumption itself is a heterogeneous phenotype that may potentially limit the generalizability of our findings on specific coffee type or preparation procedure. In addition, we are currently unable to isolate and classify genetic variants into caffeine and non-caffeine aspects of coffee given that the genetic loci heavily overlap, and future research

into the biological mechanisms of the genetic instruments is warranted when more data becomes available; until then, a potential role of micronutrients attained through coffee consumption on reduction of breast cancer risk cannot be ruled out. Another limitation was that two-sample MR assumes linearity, so we could not evaluate potential existence of non-linear associations.

## Conclusions

In summary, the results of this large MR study do not support an association of genetically predicted coffee consumption on breast cancer risk, but we cannot rule out existence of a weak association.

## Supporting information

**S1 Fig.**
(JPG)

**S2 Fig.**
(JPG)

**S3 Fig.**
(JPG)

**S4 Fig.**
(JPG)

**S5 Fig.**
(JPG)

**S6 Fig.**
(JPG)

**S1 Table. Univariate mendelian randomization analyses of coffee consumption genetic variants and breast cancer.**
(XLSX)

**S2 Table. Characteristics of genetic variants associated with coffee consumption and breast cancer overall and subtypes.**
(XLSX)

**S3 Table. SNPs associated with secondary traits using Phenoscanner (http://www. phenoscanner.medschl.cam.ac.uk/upload/).**
(XLSX)

## Acknowledgments

**Disclaimer:** Where authors are identified as personnel of the International Agency for Research on Cancer / World Health Organization, the authors alone are responsible for the views expressed in this article and they do not necessarily represent the decisions, policy or views of the International Agency for Research on Cancer / World Health Organization.

## Author Contributions

**Conceptualization:** Konstantinos K. Tsilidis.

**Data curation:** Merete Ellingjord-Dale, Michail Katsoulis, Jue-Sheng Ong, Stuart MacGregor, Benjamin Elsworth.

**Formal analysis:** Merete Ellingjord-Dale, Chew Yee.

**Funding acquisition:** Konstantinos K. Tsilidis.

**Investigation:** Merete Ellingjord-Dale, Nikos Papadimitriou, Chew Yee, Niki Dimou.

**Methodology:** Merete Ellingjord-Dale, Nikos Papadimitriou, Chew Yee, Dipender Gill, Jue-Sheng Ong, Stuart MacGregor, Benjamin Elsworth, Sarah J. Lewis, Richard M. Martin.

**Project administration:** Konstantinos K. Tsilidis.

**Resources:** Konstantinos K. Tsilidis.

**Supervision:** Konstantinos K. Tsilidis.

**Writing – original draft:** Merete Ellingjord-Dale.

**Writing – review & editing:** Merete Ellingjord-Dale, Nikos Papadimitriou, Michail Katsoulis, Niki Dimou, Dipender Gill, Dagfinn Aune, Jue-Sheng Ong, Stuart MacGregor, Benjamin Elsworth, Sarah J. Lewis, Richard M. Martin, Elio Riboli, Konstantinos K. Tsilidis.

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
