## [Decision Letter · Decision Letter 0]

6 Nov 2020

PONE-D-20-21709

Coffee consumption and risk of breast cancer: a Mendelian Randomization study

PLOS ONE

Dear Dr. Ellingjord-Dale,

Thank you for submitting your manuscript to PLOS ONE. After careful consideration, we feel that it has merit but does not fully meet PLOS ONE’s publication criteria as it currently stands. Therefore, we invite you to submit a revised version of the manuscript that addresses the points raised during the review process.

First of all I want to apologize with the authors as the peer review process takes a long time. One of the reviewer who initially accepted to review the manuscript did not eventually do the review. The manuscript has now been reviewed by two external evaluable referees, who suggested minor revisions. I am agree to, this is a well written and as per state of the art conducted investigation. The reviewers’ comments are attached below. In addition to such comments, I would ask to the authors to specify all the parameters involved in the computation of the study power (page 7), including the type I error. It’s also not clear to the reader what parameters 0.89, 0.87 and 0.80 represent, i.e. regression coefficients? Or what else?

We look forward to receiving your revised manuscript.

Kind regards,

Matteo Rota, Ph.D.

Academic Editor

PLOS ONE

Journal Requirements:

2.Thank you for including your ethics statement: 'All studies were approved by relevant institutional review boards, and all participants provided written informed consent.'   

(a) Please amend your current ethics statement to include the full name of the ethics committee/institutional review board(s) that approved your specific study.  

(b) Once you have amended this/these statement(s) in the Methods section of the manuscript, please add the same text to the “Ethics Statement” field of the submission form (via “Edit Submission”).

3.Thank you for stating the following in the Acknowledgments Section of your manuscript:

[The

breast cancer genome-wide association analyses were supported by the Government of

Canada through Genome Canada and the Canadian Institutes of Health Research, the

‘Ministère de l’Économie, de la Science et de l’Innovation du Québec’ through Genome

Québec and grant PSR-SIIRI-701, The National Institutes of Health (U19 CA148065,

18

X01HG007492), Cancer Research UK (C1287/A10118, C1287/A16563, C1287/A10710) and

The European Union (HEALTH-F2-2009-223175 and H2020 633784 and 634935).]

 [The funders had no role in study design, data collection and analysis, decision to publish, or preparation of the manuscript]

Reviewers' comments:

Reviewer's Responses to Questions

**Comments to the Author**

1. Is the manuscript technically sound, and do the data support the conclusions?

Reviewer #1: Yes

Reviewer #2: Partly

2. Has the statistical analysis been performed appropriately and rigorously? 

Reviewer #1: Yes

Reviewer #2: Yes

3. Have the authors made all data underlying the findings in their manuscript fully available?

Reviewer #1: Yes

Reviewer #2: Yes

4. Is the manuscript presented in an intelligible fashion and written in standard English?

Reviewer #1: Yes

Reviewer #2: Yes

5. Review Comments to the Author

Reviewer #1: I think this is an interesting research with large genetic data in Europe.

#1 What kinds of subtypes did you assess? I understand that you investigated the relationship between genetically predicted coffee consumption and risk of breast cancer overall as well as breast cancer subtypes incorporating several MR methods to assess the impact of potential MR assumption violations. However, there were no information about subtypes you assessed in method. There are several subtypes in breast cancer, so please state which subtypes you assessed in method section.

#2 Have you considered the dose-response analysis, especially for the postmenopausal women? There is a dose-response meta-analysis regarding the association between coffee intake and breast cancer risk. (Nutrients. 2018 Jan 23;10(2):112) The result showed that consumption of four cups of coffee per day was associated with a 10% reduction in postmenopausal cancer risk. If you have done, please include it in the results.

#3 You mentioned that you have considered some potentially important confounders of coffee and breast cancer association (BMI, age at menarche, alcohol, smoking, and age at menopause). How did you choose these factors? There are more factors potentially associated with breast cancer, as parity, age at first birth, family history of breast cancer, and use of menopausal hormone therapy. (Biochim Biophys Acta. 2015 Aug;1856(1):73-85) Is it possible to consider these factors in the analysis?

Reviewer #2: Ellingjord-Dale, M. et al performed a 2-sample Mendelian randomization analysis, accompanied by comprehensive sensitivity analyses to investigate the association between coffee intake and risk of breast cancer. Authors observed null associations in both primary and sensitivity analyses. The manuscript is well written and easy to follow. Below are my suggestions/concerns that need to be addressed in the revision.

• Given substantial heterogeneity in ratio estimates between variants, it is misleading to present results from the fixed IVW analyses. I suggest to remove “IVW fixed” results from figure 1, 2 and 3.

• In the conclusion, authors state “…, but we cannot rule out existence of weak inverse association”. I agree with authors that there could be a weak association which was not picked up by the study, but I do not think it is appropriate to infer directionality of such association.

• In the discussion, authors should also acknowledge linearity assumption with their MR analyses. If there is a threshold adverse effect, this could have been masked by fitting a linear model.

6. PLOS authors have the option to publish the peer review history of their article (what does this mean?). If published, this will include your full peer review and any attached files.

Reviewer #1: No

Reviewer #2: No

---

## [Author Response · Author response to Decision Letter 0]

13 Dec 2020

RESPONSES TO REVIEWERS -PONE-D-20-21709

Coffee consumption and risk of breast cancer: a Mendelian Randomization study

Editor:

#1 I would ask to the authors to specify all the parameters involved in the computation of the study power (page 7), including the type I error. It’s also not clear to the reader what parameters 0.89, 0.87 and 0.80 represent, i.e. regression coefficients? Or what else?

Response: We have specified the parameters in the computation of the study power under the heading Statistical power page 7, including the type I error rate.

#2 Revised financial disclosure

Response: This work was supported by the World Cancer Research Fund International Regular Grant Programme (WCRF 2014/1180 to Konstantinos K. Tsilidis). RMM was supported by a Cancer Research UK (C18281/A19169) programme grant (the Integrative Cancer Epidemiology Programme) and is part of the Medical Research Council Integrative Epidemiology Unit at the University of Bristol supported by the Medical Research Council (MC_UU_12013/1, MC_UU_12013/2, and MC_UU_12013/3) and the University of Bristol. RMM is also supported by the National Institute for Health Research (NIHR) Bristol Biomedical Research Centre which is funded by the National Institute for Health Research (NIHR) and is a partnership between University Hospitals Bristol NHS Foundation Trust and the University of Bristol.

We would be grateful if you could change the relevant online submission form on our behalf.

#3 Revised ethics statement

Response: Not applicable, as only publicly available summary association data were used.

Reviewer #1: 

#1 What kinds of subtypes did you assess? I understand that you investigated the relationship between genetically predicted coffee consumption and risk of breast cancer overall as well as breast cancer subtypes incorporating several MR methods to assess the impact of potential MR assumption violations. However, there were no information about subtypes you assessed in method. There are several subtypes in breast cancer, so please state which subtypes you assessed in method section. 

Response: Thank you. We have added more information about the estrogen receptor positive and negative breast cancer in the Methods section (Genetic data on breast cancer).

#2 Have you considered the dose-response analysis, especially for the postmenopausal women? There is a dose-response meta-analysis regarding the association between coffee intake and breast cancer risk. (Nutrients. 2018 Jan 23;10(2):112) The result showed that consumption of four cups of coffee per day was associated with a 10% reduction in postmenopausal cancer risk. If you have done, please include it in the results.

Response: We have already performed analyses per 1 cup per day of genetically predicted coffee consumption. We are not able to run analyses only on postmenopausal women since the available GWA studies on breast cancer presented information on overall breast cancer, but the majority (85%) of the included women were postmenopausal. We have acknowledged this issue in the limitation section of the Discussion.

#3 You mentioned that you have considered some potentially important confounders of coffee and breast cancer association (BMI, age at menarche, alcohol, smoking, and age at menopause). How did you choose these factors? There are more factors potentially associated with breast cancer, as parity, age at first birth, family history of breast cancer, and use of menopausal hormone therapy. (Biochim Biophys Acta. 2015 Aug;1856(1):73-85) Is it possible to consider these factors in the analysis? 

Response: We ran sensitivity analyses excluding SNPs in our genetic instrument associated with risk factors for breast cancer to probe into potential horizontal pleiotropy. The reason for not excluding SNPs associated with parity, age at first birth or HRT use was that relevant GWA studies have not been performed or the SNPs in our instrument were not associated with these risk factors.

Reviewer #2: 

#1 Given substantial heterogeneity in ratio estimates between variants, it is misleading to present results from the fixed IVW analyses. I suggest to remove “IVW fixed” results from figure 1, 2 and 3. 

Response: We have removed the fixed IVW results from figures 1, 2 and 3.

#2 In the conclusion, authors state “…, but we cannot rule out existence of weak inverse association”. I agree with authors that there could be a weak association which was not picked up by the study, but I do not think it is appropriate to infer directionality of such association. 

Response: We have removed the direction of the weak association both in the conclusion in the Abstract (page 3) and in the Conclusion of the manuscript (page 16). Thank you.

#3 In the discussion, authors should also acknowledge linearity assumption with their MR analyses. If there is a threshold adverse effect, this could have been masked by fitting a linear model.

Response: We have acknowledged the linearity assumption issue in the limitation section of the Discussion.

---

## [Decision Letter · Decision Letter 1]

4 Jan 2021

Coffee consumption and risk of breast cancer: a Mendelian Randomization study

PONE-D-20-21709R1

Dear Dr. Ellingjord-Dale,

We’re pleased to inform you that your manuscript has been judged scientifically suitable for publication and will be formally accepted for publication once it meets all outstanding technical requirements.

Kind regards,

Matteo Rota, Ph.D.

Academic Editor

PLOS ONE

Additional Editor Comments (optional):

This editor judged as resolved the comments raised by Reviewer #1. Reviewer #2 suggested to accept the manuscript for publication, too. We are pleased to inform you that your manuscript can now be published in PLOS ONE.

Reviewers' comments:

Reviewer's Responses to Questions

**Comments to the Author**

1. If the authors have adequately addressed your comments raised in a previous round of review and you feel that this manuscript is now acceptable for publication, you may indicate that here to bypass the “Comments to the Author” section, enter your conflict of interest statement in the “Confidential to Editor” section, and submit your "Accept" recommendation.

Reviewer #2: All comments have been addressed

2. Is the manuscript technically sound, and do the data support the conclusions?

Reviewer #2: Yes

3. Has the statistical analysis been performed appropriately and rigorously? 

Reviewer #2: Yes

4. Have the authors made all data underlying the findings in their manuscript fully available?

Reviewer #2: Yes

5. Is the manuscript presented in an intelligible fashion and written in standard English?

Reviewer #2: Yes

6. Review Comments to the Author

Reviewer #2: Review Comments to the Author

Please use the space provided to explain your answers to the questions above. You may also include additional comments for the author, including concerns about dual publication, research ethics, or publication ethics. (Please upload your review as an attachment if it exceeds 20,000 characters) (Limit 100 to 20000 Characters)

Authors have addressed all my concerns.

7. PLOS authors have the option to publish the peer review history of their article (what does this mean?). If published, this will include your full peer review and any attached files.

Reviewer #2: No

---

## [Editor Report · Acceptance letter]

8 Jan 2021

PONE-D-20-21709R1 

Coffee consumption and risk of breast cancer: a Mendelian Randomization study 

Dear Dr. Ellingjord-Dale:

I'm pleased to inform you that your manuscript has been deemed suitable for publication in PLOS ONE. Congratulations! Your manuscript is now with our production department. 

Kind regards, 

on behalf of

Dr. Matteo Rota 

Academic Editor

PLOS ONE